# The Preparation of Covalent Bonding COF-TpBD Coating in Arrayed Nanopores of Stainless Steel Fiber for Solid-Phase Microextraction of Polycyclic Aromatic Hydrocarbons in Water

**DOI:** 10.3390/ijerph20021393

**Published:** 2023-01-12

**Authors:** Zihan Li, Mengqi Yang, Xuetong Shen, Hongtao Zhu, Baohui Li

**Affiliations:** 1Department of Environmental Science and Engineering, North China Electric Power University, Baoding 071066, China; 2Hebei Key Lab Power Plant Flue Gas Multipollutant, Baoding 071003, China

**Keywords:** solid-phase microextraction, COF-TpBD, polycyclic aromatic hydrocarbons

## Abstract

Covalent organic framework (COF)-TpBD was grafted on the arrayed nanopores of stainless steel fiber (SSF) with (3-aminopropyl) triethoxysilane as the cross-linking agent. The prepared SSF bonded with COF-TpBD showed high thermal and chemical stability and excellent repeatability. The prepared SSF bonded with COF-TpBD was also used for the solid-phase microextraction (SPME) of seven kinds of polycyclic aromatic hydrocarbons (PAHs) in actual water samples, followed by gas chromatography with flame ionization detection (GC-FID) determination, which exhibited low limits of detection (LODs), good relative standard deviation (RSD) and high recoveries.

## 1. Introduction

Polycyclic aromatic hydrocarbons (PAHs), as a class of toxic persistent organic pollutants, not only have a serious impact on human health but also affect the water ecological environment through the biological chain [1,2,3]. Therefore, the monitoring of PAHs in environmental samples is extremely significant. Considering the low content of PAHs in the water environment, a reasonable pre-treatment is usually required before chromatographic separation.

Solid-phase microextraction (SPME) was introduced in 1990 by Pawliszyn and Arthur [4], which is a sample pre-treatment method that integrates collection, extraction, enrichment and injection into one single step. Compared with other pre-treatment methods, SPME has been widely used in medical [5,6], food [7,8], biological [9,10] and environmental [11,12] fields due to its solvent-free, recyclable processing characteristics and easy combination with chromatographic methods. The fiber coating plays a key component in SPME. Traditional commercial fibers, such as polyacrylate (PA) [13,14], poly (dimethyl siloxane) (PDMS) [15,16] and poly (dimethyl siloxane)/divinylbenzene (PDMS/DVB) [17,18] have the disadvantages of limited selectivity and fragile substrate. In order to improve the efficiency of extraction and the field of application, increasing numbers of new coating materials have been studied. So far, many materials, such as polymeric ionic liquid [19,20], graphene [21,22], Metal organic frameworks (MOFs) [23,24] and Molecular Imprinting Polymers (MIPs) [25,26] have been used as SPME coating.

Covalent organic frameworks (COFs), a porous organic material consisting of organic monomers in an orderly manner, were discovered by Yaghi in 2005 [27]. Thanks to their excellent chemical stability, adjustable pore size, low-density framework and high specific surface area [28,29,30,31], COFs were employed to various fields such as gas capture [32,33], catalysis [34,35], sensing [36,37], adsorption [38,39], extraction [40,41] and separation [42,43]. To date, all kinds of COFs have already been used in SPME [44,45,46,47]. Nevertheless, many studies only immobilized COFs on the fibers by physical coating or chemical bonding. The exfoliation of the coating is usually inevitable due to the repeated friction during the extraction procedure. Improving the stabilization of the coating is of importance for SPME. It was reported that a kind of arrayed nanoporous stainless steel fiber (SSF) with a large specific surface area was fabricated through an electrochemical etching method [48,49,50,51]. Grafting the COFs coating in the nanopores of SSF would immobilize the coating effectively, which can increase the service time of the coating.

Herein, we report the covalent bonding approach to fabricating the robust COF-TpBD coating because the coating was anchored in the nanopores of SSF. The prepared fiber was applied to establish a novel SPME method for the determination of trace PAHs in spiked lake water and river water samples by coupling with gas chromatography with flame ionization detection (GC-FID), which provide excellent stability, high reproducibility and superior extraction ability.

## 2. Materials and Methods

### 2.1. Chemicals

All chemicals used were at least of analytical grade without further purification. The 304 stainless steel fiber (220 mm/350 μm) was purchased from Shanghai Gaoge Industry and Trade Co., Ltd. (Shanghai, China). Methanol (MeOH) (≥99.9%), anhydrous ethanol (EtOH) (≥99.7%), ethylene glycol (99.0%), acetic acid (≥36.0%) and NaCl were from Tianjin Kermel Chemical Reagent Co., Ltd. (Tianjin, China). Perchloric acid (≥70.0%) was obtained from Tianjin Zhengcheng Chemical Products Co., Ltd. (Tianjin, China). Acetone (≥99.0%) was acquired from Damao Chemical Reagent Factory (Tianjin, China). Biphenyl (BIP) (≥99.0%), acenaphthylene (ACY) (≥98.0%), fluorene (FLR) (≥98.0%), anthracene (ANT) (≥98.0%), fluoranthene (FLT) (≥99.0%), pyrene (PYR) (≥98.0%), benzo[a]anthracene (BaA) (≥98.0%), (3-aminopropyl) triethoxysilane (APTES) (≥97.0%) and 1,4-dioxane (≥97.0%) were acquired from Shanghai Dibai Chemical Co., Ltd. (Shanghai, China). The mesitylene (≥97.0%), 1,3,5-Triformylphloroglucinol (Tp) (≥98.0%) and benzidine (BD) (≥97.0%) were acquired from Aladdin Chemical Co., Ltd. (Shanghai, China). Ultra-pure water collected from a Milli-Q integral system (Millipore China Co., Ltd., Shanghai, China) was used throughout the experiment.

### 2.2. Instrumentation

The GC-2014C gas chromatography was purchased from Shimadzu Co., Ltd. (Beijing, China). The CAP-1 chromatographic column was acquired from Restek (Beijing) Co., Ltd. (Beijing, China). The DP800A series programmable linear DC power supply was bought from Beijing Puyuanjingdian Technology Co., Ltd. (Beijing, China). The DF-101SB magnetic stirring oil bath was acquired from Changzhou Jintan Liangyou Instrument Co., Ltd. (Changzhou, China). The 213 Pt electrodes were obtained from Shanghai INESA Scientific Instrument Co., Ltd. (Shanghai, China). The HJ-4A digital magnetic stirrer was purchased from Shandong Oulaibo: Instrument Co., Ltd. (Jinan, China). The 5 µL syringes were from Shanghai Gaoge Industrial and Trade Co., Ltd. (Shanghai, China).

The Fourier transform-infrared (FT-IR) spectra were measured on a Tensor II (Bruker AXS GmbH, Karlsruhe, Germany). The thermogravimetric analysis (TGA) was performed on a TGA 4000 thermal gravimetric analyzer (Rigaku, Tokyo, Japan). The energy dispersive spectrometer (EDS) images and the scanning electron microscope (SEM) images were recorded on a Hitachi S4800 (Hitachi, Japan).

### 2.3. Chromatographic Condition

The high-purity nitrogen was employed as carrier gas at a flow rate of 30 mL∙min^−1^, hydrogen and air flow rates were adjusted to 40 mL∙min^−1^ and 400 mL∙min^−1^, respectively. The column was held at 120 °C for 1 min, increased to 180 °C at a rate of 20 °C∙min^−1^ and kept constant for 2 min, then increased to 250 °C at 20 °C∙min^−1^ and kept for 5 min. The injector temperature was set at 280 °C and all injections were carried out on the splitless mode for 2 min.

### 2.4. Fabrication of the Electrochemical Etched SSF

A homemade four-electrode electrochemical etching device was employed to prepare an arrayed nanoporous SSF (AN-SSF) (Figure 1) [52,53]. All of the electrodes were connected to a linear DC power supply. Firstly, the 304 SSF was polished with sandpaper, then sonicated in methanol for 15 min to remove the organic matter. After being rinsed gently with ultrapure water, the SSF was dried with nitrogen. The end of the SSF was etched in the homemade electrochemical etching device at 25 V, with a mixture of anhydrous glycol and perchloric acid (*v/v*, 9:1) as the electrolyte. After being etched for 50 s, the AN-SSF was obtained. The AN-SSF was washed with ultrapure water and dried in an oven.

### 2.5. Fabrication of the COF-TpBD Bonded Fiber

Firstly, the prepared AN-SSF was immersed into the solution of APTES/EtOH (*v/v*, 2:8) mixture at 50 °C for 8 h. The fibers were removed from the solution and immediately dried in an oven for 1 h to achieve the amino-functionalized AN-SSF. Then, 63 mg of Tp and 48 mg of BD were added to a mixture of 2.5 mL mesitylene and 2.5 mL 1-4-dioxane of ultrasonic shock with 36% acetic acid solution as the catalyst. The amino-functionalized AN-SSF was placed vertically into the fully mixed solution in the PTFE reactor. Last, the reactor was placed in a vacuum drying tank at 120 °C for 72 h. The preparation process of the COF-TpBD-bonded AN-SSF is shown in Figure 2.

### 2.6. Sample Preparation

The water samples were collected from Rixin Lake and tap water. The stock solutions were diluted to a spiked level (biphenyl, acenaphthylene and fluorene 50 μg∙L^−1^; anthracene, 30 μg∙L^−1^; fluoranthene, pyrene and benzo[a]anthracene 25 μg∙L^−1^) with different water samples.

### 2.7. SPME Procedures

The prepared COF-TpBD-bonded AN-SSF was placed into a 5 µL syringe and aged at a GC injection port at 300 °C until a stable baseline was acquired. All extractions were performed in direct extraction SPME mode. The conditioned COF-TpBD-bonded AN-SSF was immersed into the working standard solution containing 2% NaCl in an extraction vial for 30 min SPME. Then the fibers were removed from the extraction vial and immediately transferred to the GC inlet for desorption at 300 °C for 2 min.

## 3. Results and Discussion

### 3.1. Characterization of the COF-TpBD-Bonded AN-SSF

Four-electrode electrochemical etching of the bare SSF leads to arrayed nanopores on the surface of SSF (Figure 3a,b). It can be observed that there were 80–100 nm nanopores arrayed on the surface of the SSF. It is proven that COF-TpBD coating was fabricated on the surface with different magnifications (Figure 3c,d). The generated TpBD shows a particulate structure with a diameter of 35–45 nm. According to FT-IR spectra of autonomously synthesized COF-TpBD material (Figure 4a), the characteristic features were the stretching vibrations of the C-N at 1292 cm^−1^, the skeletal vibration of C=C at 1452 cm^−1^ and the stretching vibration of C=N at 1618 cm^−1^. The chemical compositions of the AN-SSF and the COF-TpBD-bonded AN-SSF coating materials were analyzed by EDS (Figure 5). By comparison, it was found that the content of C and O elements is significantly increased. The above characterization confirmed the grafting of COF-TpBD in the array microwells of SSF successfully.

The COF-TpBD coating retains good thermal stability up to 350 °C (Figure 4b), which is suitable for GC analysis. To gain the surface of AN-SSF, stainless steel sheets with the same electrochemical etching were characterized by BET (Figure 4c,d). The apparent surface area of the untreated stainless steel sheet was 1.43 × 10^−4^ m^2^∙g^−1^. In contrast, the specific surface area of the stainless steel sheet with etching was 0.423 m^2^∙g^−1^, which was a significant surface area improvement. Increased specific surface area can provide more stable anchor points for the coating and enhance the performance of the prepared fibers.

### 3.2. Optimization of Extraction and Desorption Parameters

The effects of extraction and desorption parameters including extraction time, NaCl concentration, desorption temperature and desorption time were studied in detail.

#### 3.2.1. Effect of Extraction Conditions

The effect of extraction times from 10 min to 40 min (Figure 6a) were evaluated since adequate equilibrium partitioning between stationary and aqueous phases for SPME is indispensable. The extraction efficiency of PAHs increased when extraction time changed from 10 min to 30 min. When extraction time increased further, the extraction efficiency remained unchanged. The results suggested that 30 min was sufficient to reach adsorption equilibrium. Consequently, 30 min was finally selected as the optimal extraction time.

The effect of the NaCl concentration on extraction efficiency was investigated while the NaCl concentration varied from 0% to 3% (Figure 6b). The extraction efficiency increased when the NaCl concentration changed from 0% to 2% due to the variation of solubility of non-polar organic compounds in water. However, when the NaCl concentration was above 2%, the extraction efficiency decreased significantly owing to the growth of viscosity of the solution and the salting-out effect. Therefore, the 2% NaCl Concentration was chosen in the following extractions.

#### 3.2.2. Effect of Desorption Conditions

The effect of desorption time was examined to enable complete desorption of the adsorbed analytes from the SPME fibers. However, as the extracted fibers are exposed to high temperatures for an extended time, the service life of fibers is shortened. The desorption time profiles ranging from 0.5 min to 2.5 min were studied (Figure 6c). It was found that 2 min of desorption time was adequate to achieve desorption of the analyte.

The higher desorption temperature helped with the desorption of analytes from the extracted fibers. Unfortunately, too high desorption temperatures may shorten the service time of the fibers, which depends on the thermal stability of coating. Based on the differences among the various materials, the effect of desorption temperature in the range of 260 to 300 °C was studied (Figure 6d). The extraction efficiency improved when desorption temperature varied from 260 to 300 °C. In view of the maximum permissible temperature showed by TGA, 300 °C was selected for the subsequent studies.

### 3.3. Analytical Figures of Merit

The chromatogram of seven kinds of PAHs in optimized conditions is shown in Figure 7. Table 1 summarizes the analytical performance of the COF-TpBD-bonded AN-SSF for the determination of PAHs by GC-FID, including linear range (LR), correlation coefficient (R), limits of detection (LODs), relative standard deviation (RSD) and the recovery rate. The developed method provided the LR of 25–1000 µg·L^−1^ for BIP, ACY and FLR, 10–1000 µg·L^−1^ for ANT and 5–1000 µg·L^−1^ for FLT, PYR and BaA. All the PAHs exhibit wide linear ranges with good linearity (R^2^ ˃ 0.9935). The LODs were in the range of 1.00–5.00 µg·L^−1^. Single-fiber reproducibility was gained by using the COF-TpBD-bonded AN-SSF to detect PAHs in stock solutions. The relative standard deviations (RSDs) for intra-day were 1.7–9.0%. Furthermore, the RSDs among the three parallel-prepared fibers were in the range of 2.1–7.2%. The spiked recovery rates were 87.59–162.92%. The results for the actual water samples without spiking are not detected. The fiber extraction efficiency did not change significantly over 150 cycles of the experiment. The results show that the COF-TpBD-bonded AN-SSF prepared by this method has good repeatability and stability.

## 4. Conclusions

In summary, we report a method for covalently immobilizing COF-TpBD in the nanopores of AN-SSF. The prepared COF-TpBD-bonded AN-SSF was employed to develop a new SPME-GC-FID to determinate trace PAHs in water samples. The developed analytical method provided LODs of 1.0–5.0 µg·L^−1^ for seven kinds of PAHs. The precision of the developed method with a single fiber over a six-day period ranged from 1.7% to 9.0%. The reproducibility between three fibers prepared simultaneously by the same method ranged from 2.1% to 7.2%. One COF-TpBD bonded fiber can withstand at least 150 cycles without significant efficiency loss in extraction. The recoveries for actual spiked water samples were 87.59–162.92%. The method has excellent feasibility for the determination of trace PAHs. The large surface area, suitable pore size and aromatic functionality of COF-TpBD endow it with great potential for the preconcentration of aromatic toxic organic pollutants in environmental samples.

## Figures and Tables

**Figure 1 ijerph-20-01393-f001:**
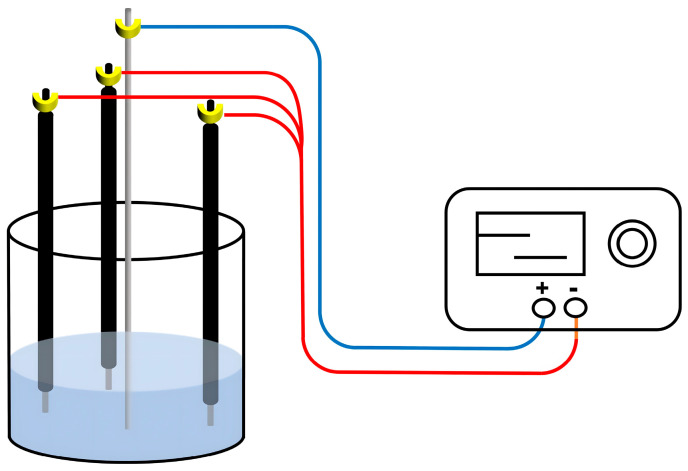
Schematic diagram of electrochemical etching device.

**Figure 2 ijerph-20-01393-f002:**
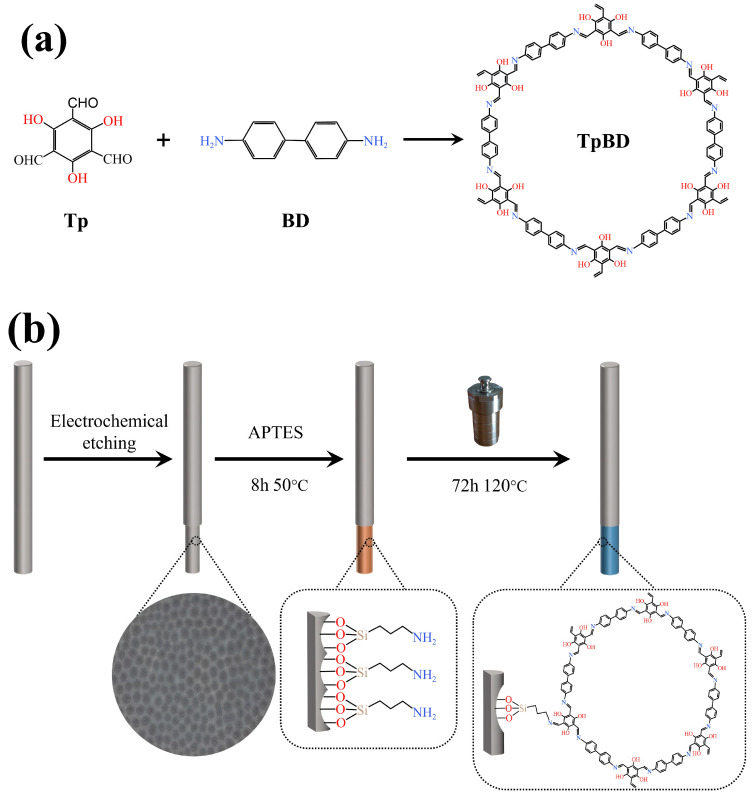
(**a**) Diagrammatic representation of the synthesis of TpBD by condensation of TP and BD. (**b**) Schematic illustration of the fabrication processes of COF-TpBD-bonded AN-SSF.

**Figure 3 ijerph-20-01393-f003:**
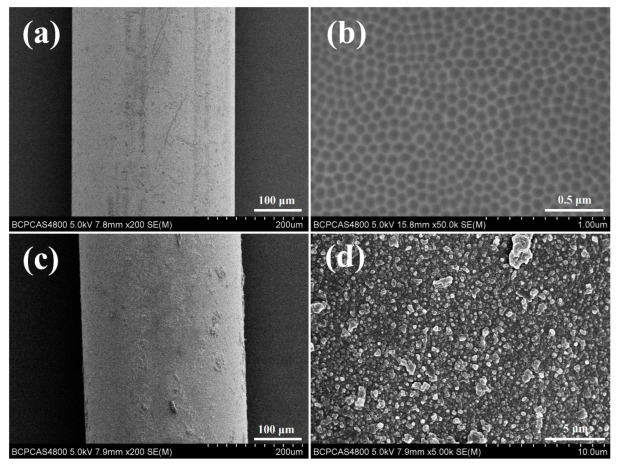
SEM images of AN-SSF with magnification at (**a**) 200× and (**b**) 50,000×; SEM images of the COF-TpBD-bonded AN-SSF with magnification at (**c**) 200× and (**d**) 5000×.

**Figure 4 ijerph-20-01393-f004:**
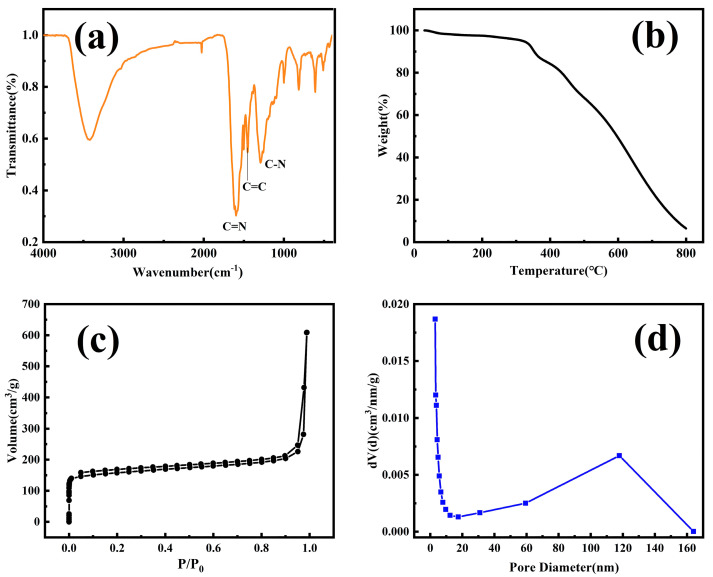
(**a**) FT-IR spectra of powder COF-TpBD; (**b**) TGA curve of powder COF-TpBD; (**c**) N_2_ adsorption–desorption isotherms of the etched stainless steel sheet; (**d**) pore size distribution of the etched stainless steel sheet.

**Figure 5 ijerph-20-01393-f005:**
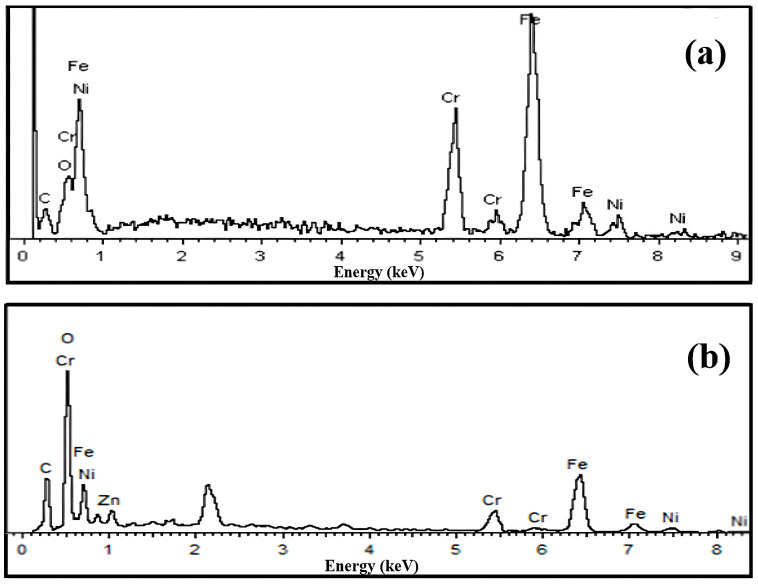
EDS spectra of the AN-SSF (**a**) and the COF-TpBD-bonded AN-SSF (**b**).

**Figure 6 ijerph-20-01393-f006:**
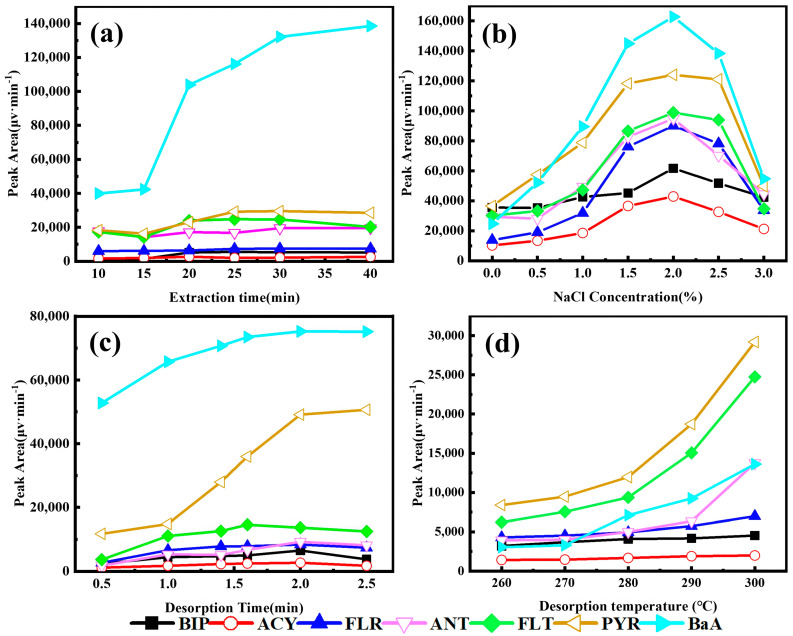
Effects of extraction and desorption parameters on the extraction efficiency for PAHs: (**a**) Extraction time (conditions: Desorption Time, 2 min; Desorption temperature, 300 °C; NaCl Concentration, 2%). (**b**) NaCl Concentration (conditions: Extraction time, 30 min; Desorption Time, 2 min; Desorption temperature, 300 °C). (**c**) Desorption Time (conditions: Extraction time, 30 min; Desorption temperature, 300 °C; NaCl Concentration, 2%). (**d**) Desorption temperature (conditions: Extraction time, 30 min; Desorption Time, 2 min; NaCl Concentration, 2%).

**Figure 7 ijerph-20-01393-f007:**
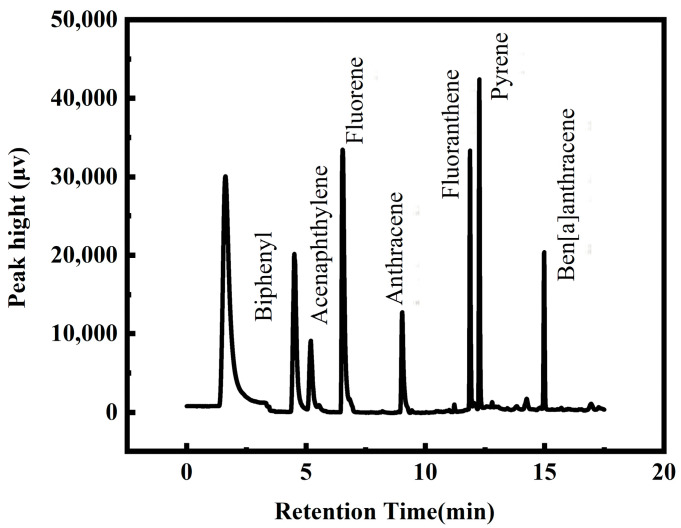
The chromatogram of seven kinds of PAHs in optimized conditions.

**Table 1 ijerph-20-01393-t001:** Analytical performance of the COF-TpBD-bonded AN-SSF for the SPME of PAHs.

Analyte	Linear Range (μg L^−1^)	R^2^	LODs (μg L^−1^)	RSD		Recovery Rate (%)
				Intra-Day (*n* = 6) (%)	Fiber-to-Fiber (*n* = 3) (%)	
BIP	25–1000	0.9988	5.00	1.9	2.1	162.92
ACY	25–1000	0.9862	5.00	7.4	3.6	155.47
FLR	25–1000	0.9938	5.00	1.7	7.0	128.13
ANT	10–1000	0.9930	3.00	8.0	7.1	101.74
FLT	5–1000	0.9878	1.00	4.0	6.4	91.91
PYR	5–1000	0.9935	1.00	3.8	7.2	145.44
BaA	5–1000	0.9890	1.00	9.0	3.6	87.59

## Data Availability

Not applicable.

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
