# Peer review of "The Preparation of Covalent Bonding COF-TpBD Coating in Arrayed Nanopores of Stainless Steel Fiber for Solid-Phase Microextraction of Polycyclic Aromatic Hydrocarbons in Water"

_ijerph, 2023, doi:10.3390/ijerph20021393_

Round 1

Reviewer 1 Report

Title: The preparation of covalent bonding COF-TpBD coating in 2 arrayed nanopores of stainless steel fiber for solid-phase 3 microextraction of polycyclic aromatic hydrocarbons in Water

In the present work, Covalent organic frameworks (COF)-TpBD was grafted on the arrayed nanopores of stain- less steel fiber (SSF) while (3-aminopropyl) triethoxysilane as the cross-linking agent. The prepared  SSF bonded with COF-TpBD was employed to develop a new SPME-gas chromatography with  flame ionization detection (GC-FID) to determinate trace polycyclic aromatic hydrocarbons (PAHs) in water samples. The developed analytical method provided the limits of detections (LODs) of 1.0- 5.0 µg·L−1 for 7 kinds of PAHs. Here, some interesting results were reported. The writing of this manuscript is acceptable. The main results are clearly presented. I think the manuscript can be accepted for publication after addressing the following concerns.

1.      The submitted manuscript and its related principles are attractive however it suffered from the lack of relevant update references. Add some recent literature to strengthen your discussion.

2.      TGA and BET results look wrong. So, the results must be repeated to check the reproducibility.

3.      The quality of all figures must be improved.

4.      Did the authors check the applicability of the method in the real samples?

5.      How would this research work advance the previous work done in the existing field of study and/or across other fields?

6.      Conclusions: this section should be different from the abstract. More specific data rather than the general description should be presented.

Reviewer 2 Report

2.7: The author should do the quality control and quality assurance. It is very important for a new analytical method. 

3.3: The authors did the standard curve for this material. But it usually uses r2 to express the linearity of a calibration curve. The recoveries of some PAHs were higher. The author should explain it.

 To validate the new material, it is also necessary to test real samples without spiking PAHs.

Round 2

Reviewer 2 Report

I have no comments on the revised manuscript.